# Changes in data management contribute to temporal variation in gestational duration distribution in the Swedish Medical Birth Registry

**Dominika Modzelewska**[1]*, **Pol Sole-Navais**[1], **Anna Sandstrom**[2,3,4], **Ge Zhang**[5,6], **Louis J. Muglia**[5,6,7], **Christopher Flatley**[8], **Staffan Nilsson**[9], **Bo Jacobsson**[1,8,10]

1 Department of Obstetrics and Gynecology, Institute of Clinical Sciences, Sahlgrenska Academy, University of Gothenburg, Gothenburg, Sweden, 2 Department of Medicine, Solna, Clinical Epidemiology Division, Karolinska Institute, Stockholm, Sweden, 3 Department of Women's and Children's health, Uppsala University, Uppsala, Sweden, 4 Department of Obstetrics and Gynecology, Oregon Health and Science University, Portland, Oregon, United States of America, 5 Division of Human Genetics, Cincinnati Children's Hospital Medical Center, Department of Pediatrics, University of Cincinnati College of Medicine, Cincinnati, Ohio, United States of America, 6 Center for Prevention of Preterm Birth, Cincinnati Children's Hospital Medical Center, Cincinnati, Ohio, United States of America, 7 Office of the President, Burroughs Wellcome Fund, Research Triangle Park, North Carolina, United States of America, 8 Department of Genetics and Bioinformatics, Division of Health Data and Digitalization, Norwegian Institute of Public Health, Oslo, Norway, 9 Department of Mathematical Sciences, Chalmers University of Technology, Gothenburg, Sweden, 10 Department of Obstetrics and Gynecology, Sahlgrenska University Hospital, Gothenburg, Region Västra Götaland, Sweden

* dominika.modzelewska@gu.se

**Data Availability Statement:** The Swedish Medical Birth Registry is a national dataset; therefore, it is considered as public property. However, access to

## Abstract

Multiple factors contribute to gestational duration variability. Understanding the sources of variability allows to design better association studies and assess public health measures. Here, we aimed to assess geographical and temporal changes in the determination of gestational duration and its reporting in Sweden between 1973 and 2012. Singleton live births between 1973 and 2012 were retrieved from the Swedish Medical Birth Register. Gestational duration trends in percentiles and rates of pre- and post-term deliveries were analyzed by plotting the values over time. Temporal changes in gestational duration based on ultrasound and last menstrual period (LMP) estimation methods were compared. Intervals between LMP date and LMP-based due date were analyzed to assess changes in expected gestational duration. In total, 3 940 577 pregnancies were included. From 1973 until 1985, the median of gestational duration estimated based on LMP or ultrasound decreased from 283 to 278 days, and remained stable until 2012. The distribution was relatively stable when ultrasound-based estimates were used. Until the mid-1990s, there was a higher incidence than expected of births occurring on every seventh gestational day from day 157 onward. On an average, these gestational durations were reported 1.8 times more often than adjacent durations. Until 1989, the most common expected gestational duration was 280 days, and thereafter, it was 279 days. The expected gestational duration varied from 279 to 281 days across different Swedish counties. During leap years, the expected gestational duration was one day longer. Consequently, leap years were also associated with significantly

the data is given only to the researches with permission from a Swedish regional ethical review board and after approval of the research plan by the data manager. The data request may be sent to The Swedish National Board of Health and Welfare (https://www.socialstyrelsen.se/).

**Funding:** BJ: 1. Swedish government grants to researchers in the public health sector, grants no.: ALFGBG-717501, ALFGBG-507701, ALFGBG-426411, URL: https://www.vr.se 2. The Swedish Research Council, grants no. 2015-02559, URL: https://www.vr.se, 3. The March of Dimes Foundation, grant no.: 21-FY16-121, URL: https://www.marchofdimes.org 4. The Burroughs Wellcome Fund Preterm Birth Research Grant, grants no.: 10172896, URL: https://www.bwfund.org LM: 1. The March of Dimes Prematurity Research Center Ohio Collaborative, URL: https://www.marchofdimes.org The funders had no role in study design, data collection and analysis, decision to publish, or preparation of the manuscript.

**Competing interests:** The authors have declared that no competing interests exist.

higher preterm and lower post-term delivery rates than non-leap years. Changes in data handling and obstetrical practices over the years contribute to gestational duration variation. The resulting increase in variability might reduce precision in association studies and hamper the assessment of public health measures aimed to improve pregnancy outcomes.

## Introduction

Gestational age at birth is one of the most important factors in predicting pregnancy and neonatal outcomes [1, 2], but the biological mechanism that initiates parturition remains unclear. Relatively large estimates of heritability, i.e., up to 31% for gestational duration and 36% for preterm delivery (PTD), suggest the importance of genetic factors [3–7]. However, it has been difficult to find specific genetic variants associated with gestational duration. The largest genome-wide association study on gestational duration performed to date (n = 43 968) discovered six loci, but these account for only 1% of population variance [8]. Larger sample sizes are required to detect smaller effects that are spread across hundreds of genetic variants. However, aspects other than sample size may account for the difficulty in discovering and replicating genetic associations.

Gestational duration estimation methods, reporting protocols, and obstetrical practices have changed globally over time, contributing to gestational duration variability. Furthermore, these changes occur at different times in different countries, and even within countries [9–12]. In the USA, an increase in the PTD rate has been reported in relation to an increase in the proportion of multiple gestations [13]; this could be due to the increased use of assisted reproductive technologies [12]. In the UK, differences in gestational age reporting (i.e. rounding gestational age to the closest week) has hampered the PTD rate comparisons between different hospitals and regions [13]. In Sweden, changes in PTD rates over time and across the country have been reported [10, 11, 14, 15]. In the first years of 1980s, the PTD rate increased with respect to a shift in the estimation method, which is from using the last menstrual period (LMP) to ultrasound-based dating [10, 16]. Since 1985 to 2001, PTD rates dropped owing to a decrease in the prevalence of deliveries between 34 and 37 completed weeks of gestation [10]. Morken et al. evaluated whether the changes in gestational duration distribution were affected by changes in the prevalence of PTD risk factors, such as maternal age, smoking, and primiparity, but no such association was found [10].

In this paper, we aim to assess the changes in determination of gestational duration and the impact that data recording practices have on accuracy. We also investigate evolving obstetric practices and their relationship with gestational duration in Sweden between 1973 and 2012.

## Materials and methods

### Sample

This study is based on the national Swedish Medical Birth Register (MBR) from 1973 to 2012. The MBR contains information on approximately 99% of births occurring in Sweden, compiling reproductive history, complications during pregnancy, delivery, neonatal period, as well as demographic information [17]. Regarding gestational duration, MBR contains LMP date, due date based on LMP date and clinical investigation, and due date based on ultrasound scan. The MBR also provides the "best estimate" for gestational duration which is based on both

availability of data related to gestational duration and the current consensus on higher accuracy of ultrasound, compared to LMP-based estimates.

The study sample was restricted to live singleton births with available gestational duration estimates in the range of 154–301 days. In line with the aim of the study, the cohort was further restricted to pregnancies with available information regarding gestational duration estimation method (LMP or ultrasound) and onset of delivery (spontaneous, cesarean section, or induction).

## Variable definitions

Analyses were based on the best gestational duration estimate listed in the MBR. The best estimate includes gestational duration retrieved from a combination of various sources and is ordered on the basis of reliability of the estimate made from such combinations [18]. In this study, the best estimates, based either on LMP (categories two, four, eight, nine, or ten of the MBR's best gestational duration estimate) or ultrasound (categories one, five, six, or seven of the MBR's best gestational duration estimate) dating, were used to define the duration of gestation (LMP- or ultrasound, respectively), as per the hierarchical set of rules laid out in the Swedish MBR [18].

Onset of labor was categorized as spontaneous or iatrogenic (induction or cesarean section). If onset of labor was recorded as "spontaneous" in a check-box, or had the International Classification of Diseases version 10th (ICD-10) codes, namely, O42 (pre-labor rupture of membranes), O75.6 (delayed delivery after spontaneous or unspecified rupture of membranes), O60.1 (preterm spontaneous labor with preterm delivery), or O60.2 (preterm spontaneous labor with term delivery) in the MBR, onset of labor was defined as spontaneous. Onset of labor was defined as iatrogenic if it was recorded in check-boxes as "induced," "started with a cesarean section," "planned cesarean section," or had the ICD-10 code O61.0 (failed medical induction of labor) in the MBR.

## Statistical analyses

To detect possible changes in the estimation or data reporting methods, temporal changes in gestational duration distribution were explored graphically. Gestational duration trends in percentiles, rates of PTD ($< 259$ gestational days) and post-term delivery ($> 294$ gestational days), and expected gestational duration were studied by plotting the values over time. To assess changes in expected gestational duration, we analyzed changes in the interval between LMP date and predicted LMP-based due date temporally and across Swedish counties. For each pregnancy, an interval was calculated by subtracting the LMP date from the LMP-based due date. This definition of expected gestational duration allowed us to not only assess changes in expected duration, but also observe whether mistakes were introduced when estimating the due date. Chi-square test of independence was used to analyze significance of temporal changes in PTD and post-term delivery rates, and between leap and non-leap years. Histograms were used to visualize unexpected frequencies of gestational duration. The occurrence of gestational durations of unexpectedly high frequency was determined by estimating the ratio of observed to expected gestational duration frequency. All analyses were performed with R software, version 3.5.1.

The study was approved by Regional Ethic Committee of the Western Health Care Region in Sweden (Dnr. 576–13). MBR is a national population-wide database; therefore, no informed consent is required. Individual-level data are anonymous. Personal identification numbers are kept and known only to the National Board of Health and Welfare.

## Results

### Gestational duration distribution

The study sample consisted of 3 940 577 pregnancies that met the inclusion criteria. During the study period, we observed several changes in gestational duration distribution. From the start of the study period until 1985, the median gestational duration decreased from 283 to 278 days, and then it plateaued until the end of the study period (Fig 1).

The drop in median gestational duration occurred with a simultaneous increase in PTD and decrease in post-term delivery rates (Fig 2). The change in PTD and post-term delivery rates correlated with the introduction of ultrasound-based estimation of gestational duration in 1982. Before 1980, the PTD rate was at its lowest, and then it gradually increased to a peak of 5.6% in 1984 (Fig 2). A steady decline followed thereafter with the incidence stabilizing at 4.8% after the 1990s (chi-square test, p < 0.01). From 1973 to 1984, post-term birth incidence dropped substantially from a high of 12.3% to 5.5% (chi-square test, p < 0.01). In the later years, post-term delivery rate fluctuated, peaking at 6% from 2001 to 2003, before declining to 4.9% in 2012.

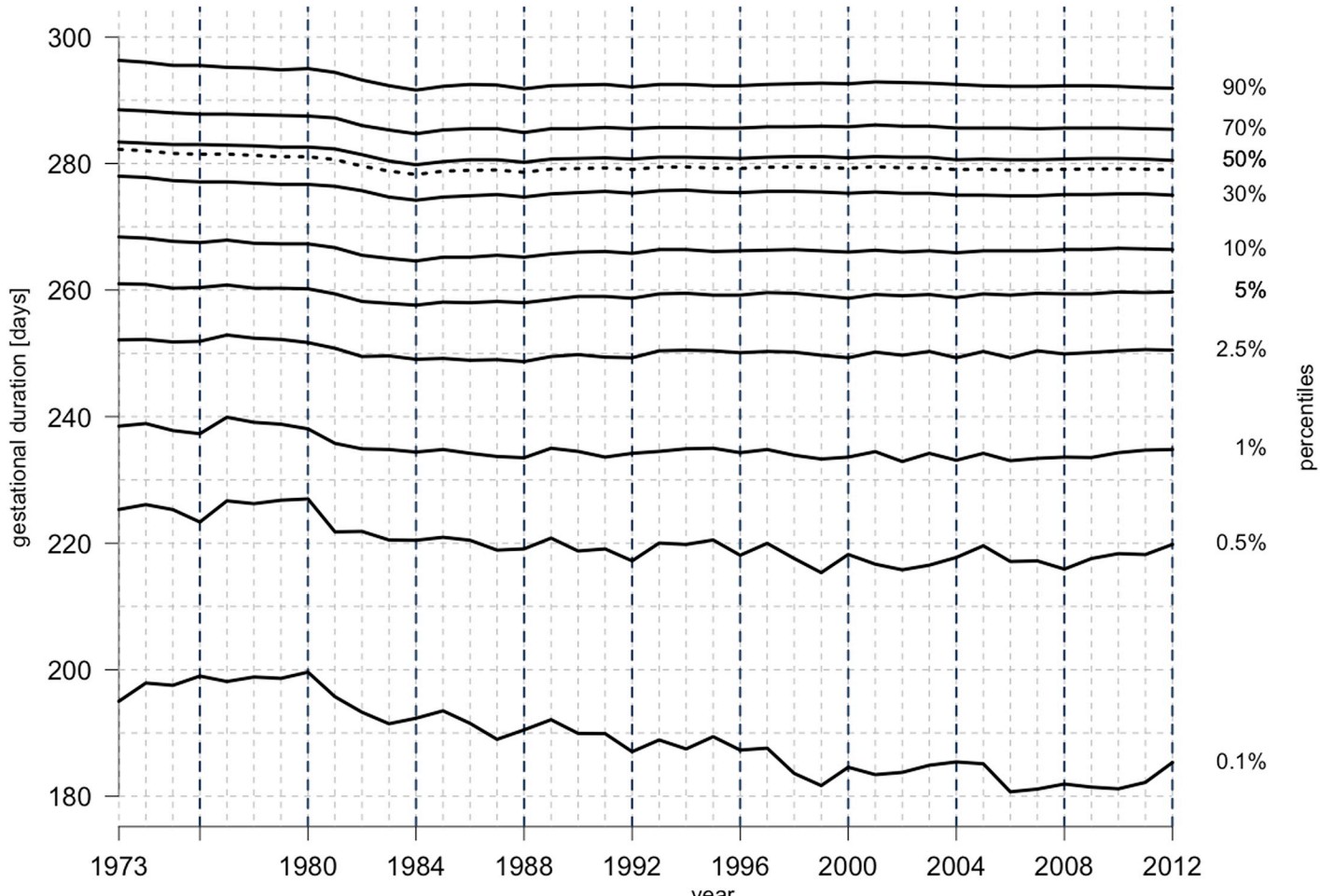

**Fig 1. Pregnancy duration distribution, Swedish Medical Birth Register, 1973–2012.** Gestational duration percentiles: 0.1th, 0.5th, 1st, 2.5th, 5th, 10th, 30th, 50th, 70th, and 90th; percentiles are marked by solid lines. Dotted line represents mean gestational duration. The left-hand y-axis indicates pregnancy duration in days and the right-hand y-axis indicates percentiles. Vertical black dashed line indicates leap years. Sample size: n = 3 940 577.

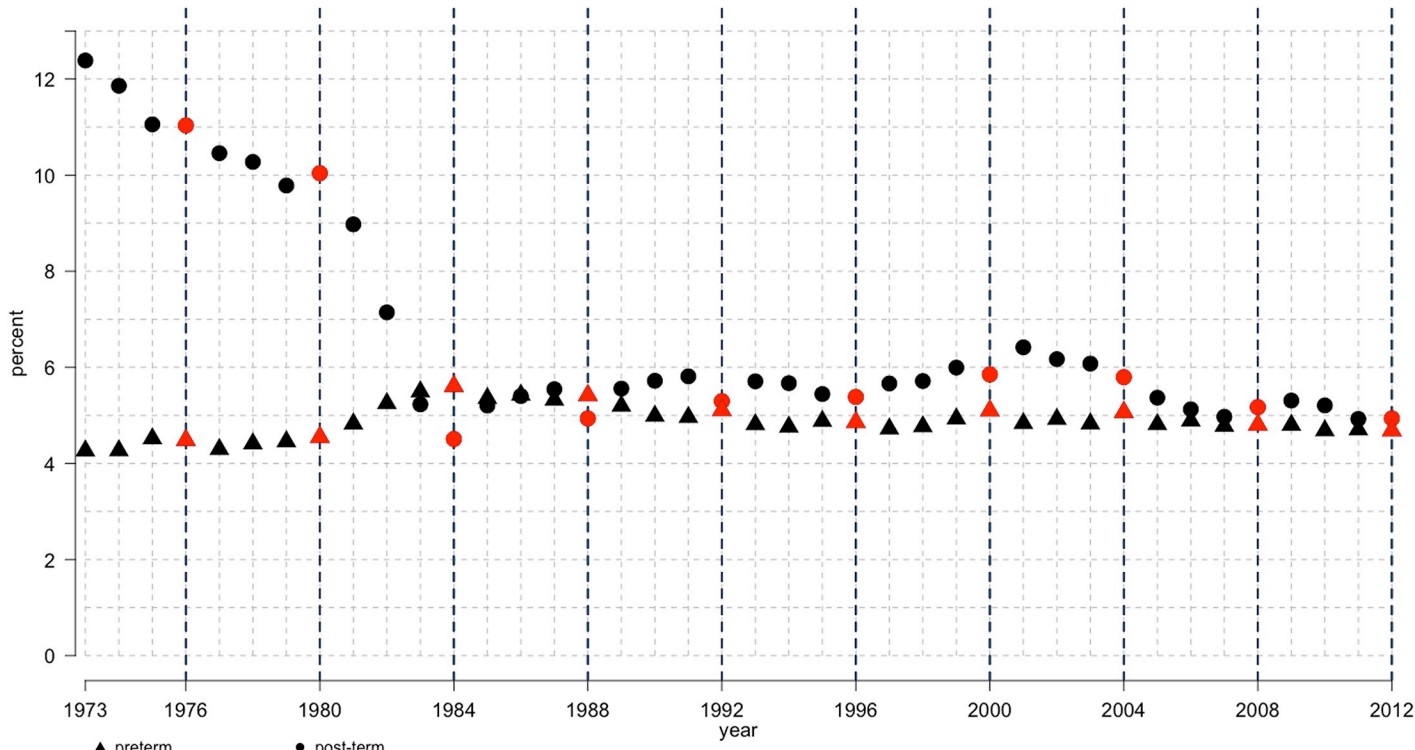

**Fig 2. Incidence of preterm and post-term deliveries, Swedish Medical Birth Register, 1973–2012.** Percentages of preterm (triangle) and post-term (circle) deliveries registered from 1973 to 2012. Red dots and vertical black dashed line marks leap years. Preterm and post-term deliveries defined as pregnancies lasting < 259 days and > 294 days, respectively. Sample size: n = 3 940 577.

Until the mid-1990s, gestational duration was commonly reported in weeks instead of days. Conversion to day-units involved multiplication of gestation duration in weeks by seven and addition of three days. This resulted in a higher incidence of births than expected, occurring on every seventh gestational day from day 157 onwards (Fig 3A). On average, these gestational duration values were reported 1.8 times more often than for the neighboring durations (Fig 3B). In 1982, the frequency of every seventh day (from day 157 onwards) was the highest (2.7 times more than expected).

## Gestational duration distribution in deliveries with spontaneous onset

To understand whether obstetric care changes had an impact on gestational duration variability, a restricted analysis of deliveries with spontaneous-onset stratified by gestational duration estimation method was conducted. Data enabling the retrieval of spontaneous-onset deliveries were only available from 1990 to 2012 period. In the distribution of ultrasound-based gestational duration, small changes were observed within the 0.1[th] percentile (Fig 4). LMP-based gestational duration distribution underwent a gradual left-shift within the 90[th] percentile, and in the lower 1[st], 0.5[th] and 0.1[th] percentiles (Fig 4).

**Gestational duration distribution in iatrogenic-onset deliveries.** Due to limitations in data availability, the distribution of gestational duration in deliveries with cesarean section or induction onsets could be studied only from 1990 to 2012 and 1999 to 2012, respectively. There was an increase in the proportion of planned cesarean sections from 4.6% in 1990 to

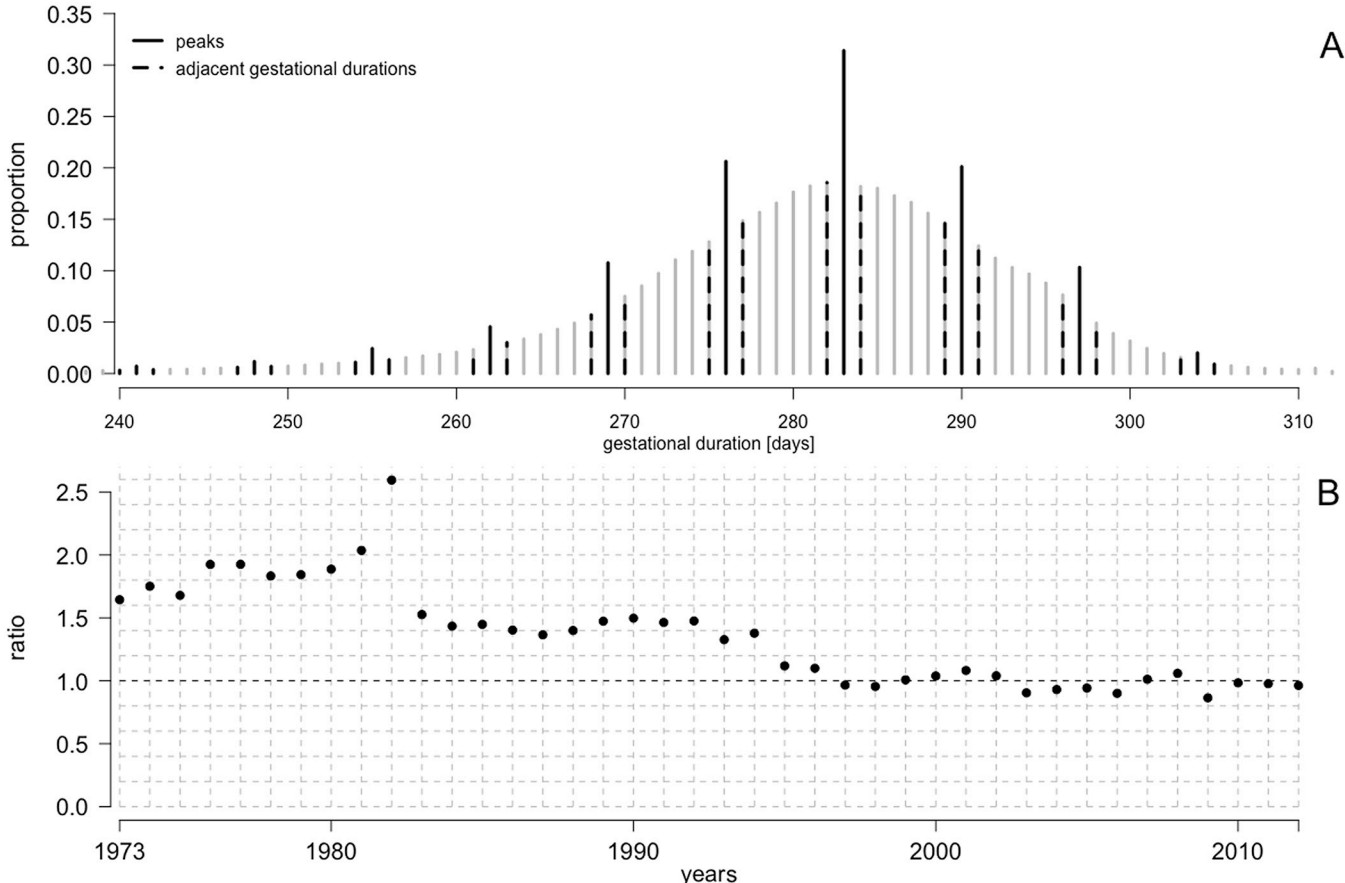

**Fig 3. Gestational duration distribution, Swedish Medical Birth Register, 1973–2012. A)** Gestational duration distribution. Gestational durations that occurred at higher frequency than expected (peaks) are marked by solid black lines. Sample was reduced to pregnancies registered from 1973 to 1982, that is, to the period in which peaks were most common. Adjacent gestational durations to peaks are marked by black dashed lines. **B)** Average ratio of peaks' frequency to the mean of adjacent gestational duration frequency from 1973 to 2012. Sample size: n = 3 940 577.

8.5% in 2012, and in the proportion of inductions from 9.3% in 1999 to 14.3% in 2012 (Fig 5A). The gestational duration distribution remained relatively constant over the years in pregnancies with planned cesarean section or with induced labor (Fig 5B).

## Changes in expected gestational duration and due date estimation

Until 1989, the most common expected gestational duration was 280 days. Thereafter, it was more common to add 279 days to the LMP date to estimate the due date (Fig 6). Until 2008, the expected gestational duration was often one day longer in leap years compared to other years (Fig 6). This extra day was due to the manual calendar-based calculation of the due date, which ignored the leap day of February 29. Overestimation of the due date led to an underestimation of gestational duration, and consequently, to an increase in PTD and decrease in post-term delivery rate. We observed peaks in PTD rates in the leap years from 1984 to 2004 (chi-square test, $p < 0.01$) (Fig 2). Post-term delivery rates decreased in the leap years from 1984 to 1992 and in 2000 (chi-square test, $p < 0.01$) (Fig 2).

Before 2002, extra variation in expected gestation duration was noticed in pregnancies with deliveries in the December–February period (i.e. women with LMP in the March–May range).

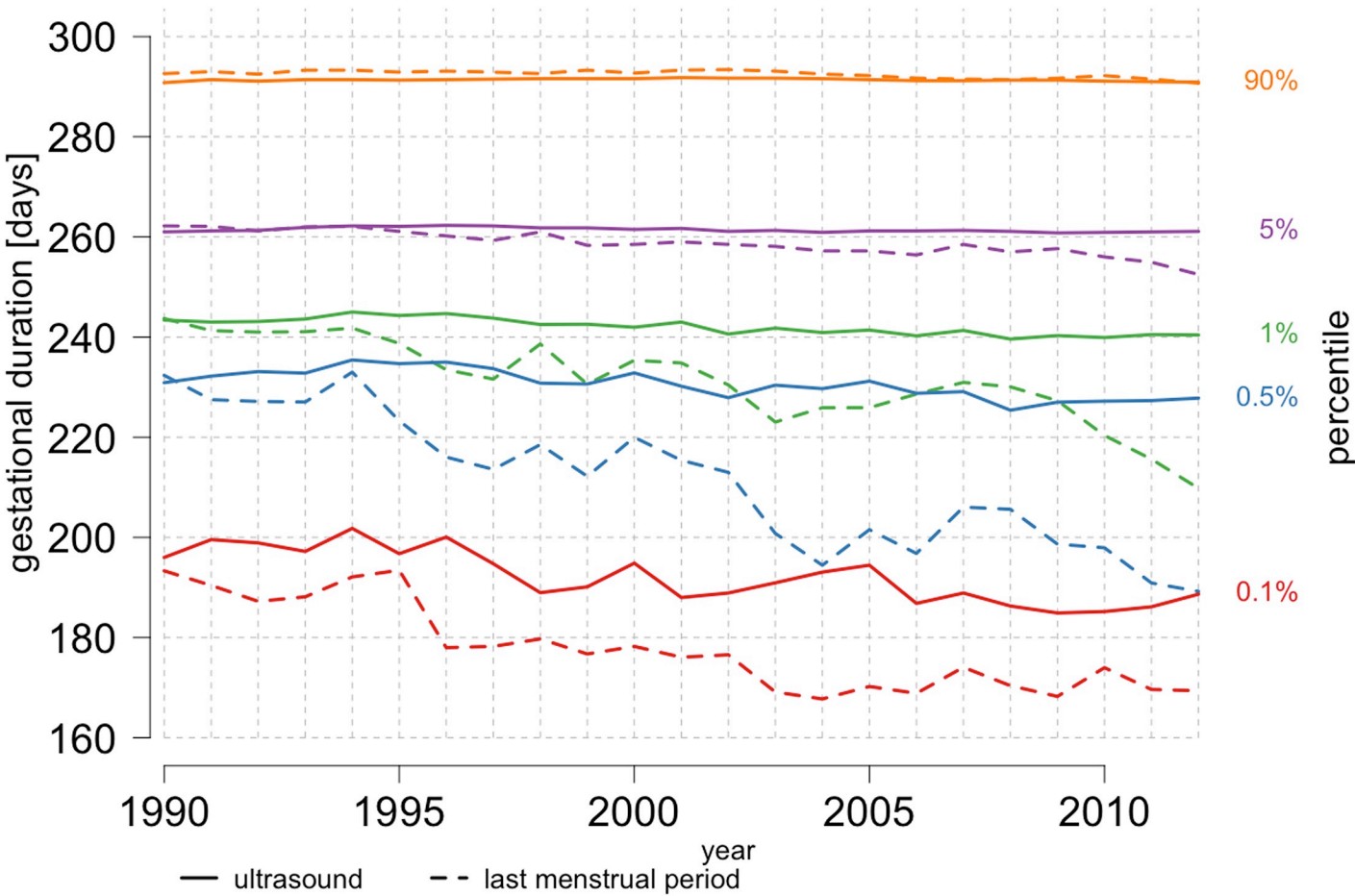

**Fig 4. Distribution of gestational duration in women with spontaneous-onset deliveries, Swedish Medical Birth Register, 1990–2012.** Temporal changes in the gestational duration percentiles: 0.1[th], 0.5[th], 1[st], 5[th], 90[th], with regard to estimation method, LMP-based (dashed line) or ultrasound-based (solid line); percentiles are respectively marked with different colors. The left-hand y-axis indicates gestational duration in days and the right-hand y-axis indicates percentiles. Sample sizes: pregnancies with LMP-based duration estimate: n = 272 416; pregnancies with ultrasound-based duration estimate: n = 1 551 133.

Women who gave birth in those months often had longer than expected gestation durations. This was because of the procedure used to estimate due date, i.e. adding 9 months 7 days to the LMP date and not 279 or 280 days. For example, if the mother's LMP date was 1 March 1985, addition of 9 months and 7 days would make her due date 8 December 1985. Therefore, the interval between due date and LMP date was longer at 282 days. Additionally, the expected gestation duration varied from 279 to 281 days across different Swedish counties (S1 Table).

## Discussion

In this study, we analyzed changes in gestational distribution, proportion of iatrogenic deliveries and changes made to the estimation of gestational duration over time. We observed unequivocal patterns in the variation of gestational duration for specific years or across the whole study period. Changes in date estimation procedures and clinical management over time underlie these patterns. In this paper, we observed several technical factors contributing to gestational duration variability. These are related to temporal and spatial variability in expected gestational duration, changes in the selection of measurement units (weeks/days) for

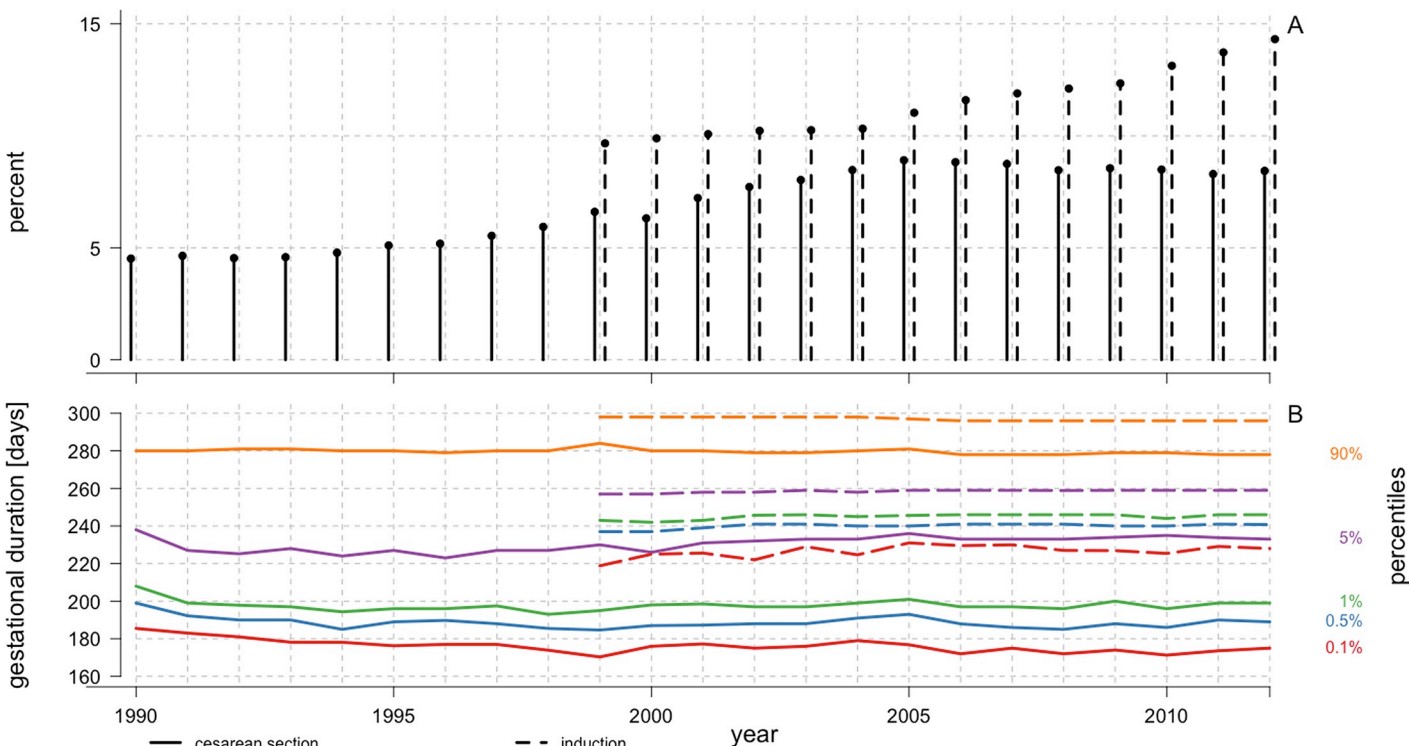

**Fig 5. Prevalence of iatrogenic onset deliveries, Swedish Medical Birth Register, 1990–2012. A)** Percentage of iatrogenic-onset delivery with regard to the type of medical intervention, i.e. cesarean section (solid line) or induction (dashed line). **B)** Gestational duration by percentiles (0.1th, 0.5th, 1st, 5th, and 90th percentiles, respectively marked by different colors in the cesarean section [solid line] and labor induction [dashed line] cohorts). Due to limited data availability, the cesarean section and induced labor cohorts were restricted between 1990 and 2012 and 1999 and 2012; n = 155 647 and 157 191, respectively.

estimation, changes in the reporting of due dates in medical records, or a lack of accounting for an extra day in leap years when estimating due dates. These differences may not have a strong impact in clinical practice but might hamper the assessment of public health measures or might affect association studies; for example, reporting gestational duration in various unit-measures (days or weeks) might decrease the correlation between the relatives, in consequence, that might affect heritability estimates.

The digitization of medical record data entry had a substantial impact on the variability of gestational duration, providing estimates that are more accurate. Electronic systems have helped reduce variation in the definition of expected gestational duration. We show that before the 1990s, the most common expected gestational duration was 280 days and thereafter, it was 279 days. However, there was never a universal definition of the expected duration of gestation and it varied temporally and geographically across Sweden. The lack of consensus on expected gestation duration was already reported in the 1990s [19]. According to a previous study, the incidence of post-term deliveries decreased by 1.7% when expected gestational duration was assumed to be 282 instead of 280 days [20]. Currently, there are three different medical record systems in Sweden; the most common, used in 17 regions, is Obstetrix®, and the others are Partus® and Cosmic® [21]. The systems differ when it comes to expected gestational duration, which is 279 days in Obstetrix® and Partus® and 280 days in Cosmic® [21].

Until around 1995, we noted a regularly high birth frequency on every seventh gestational day from day 157 onwards, suggesting that until the mid-1990s, hospitals commonly recorded gestational duration in weeks instead of days. At the registry level, this information was

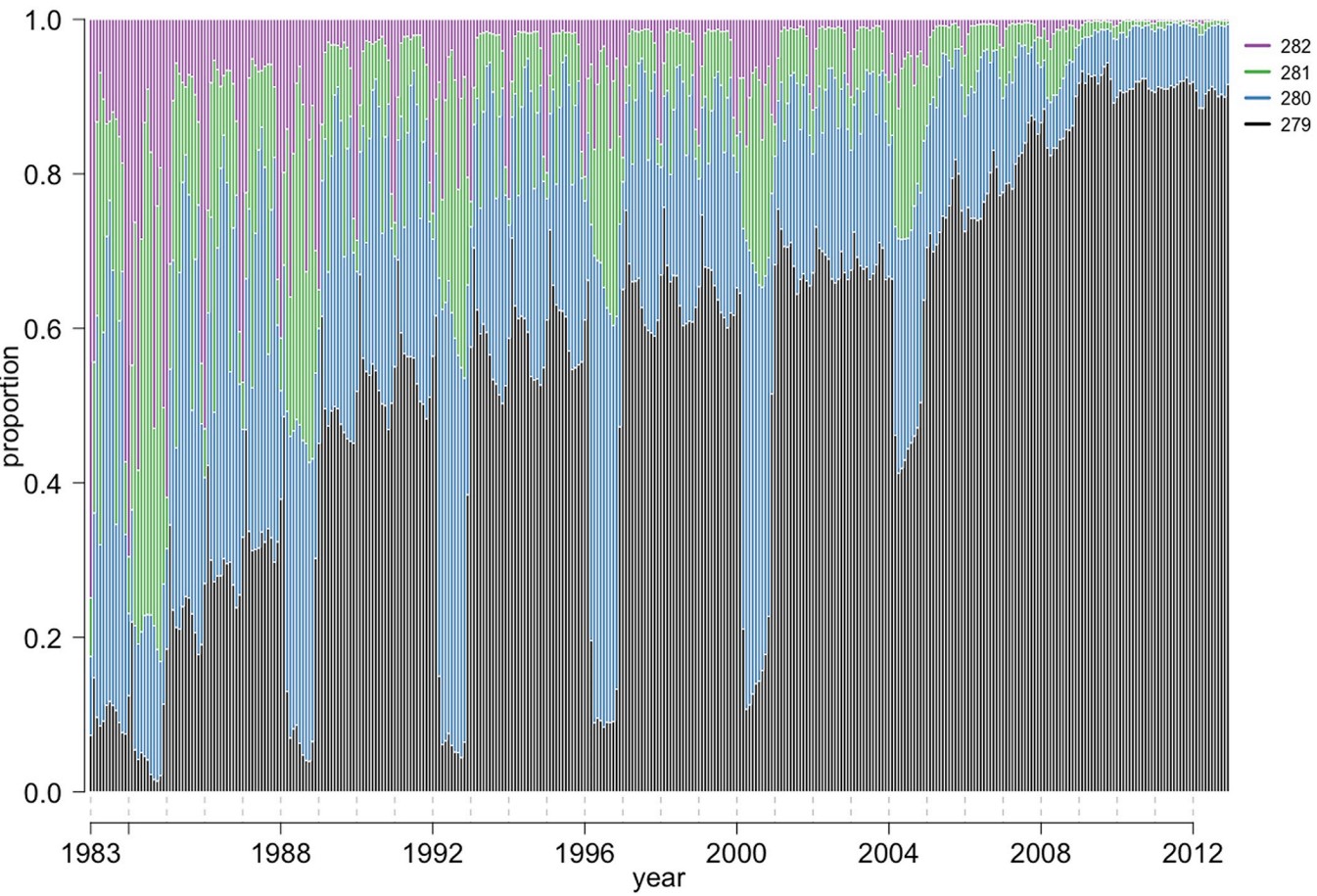

**Fig 6. Interval (days) between the LMP and due date for babies born from 1983 to 2012, Swedish Medical Birth Register, 1983–2012.** Most common intervals in days (from 279 to 282 days), between LMP and LMP-based due date for a baby born in a given month during 1983 to 2012. Sample size: n = 2 480 726.

converted into day-units by multiplying gestation duration by seven and adding three days. Such a formula was possibly used to represent the week by its middle day to provide the most optimal estimate. In 1982, the difference between the observed and expected frequency of gestational duration values was the largest. We speculate that this arose from the digitization of medical data transfer from hospitals to the MBR.

Several changes in the approach used to estimate gestational duration can be detected by analyzing the patterns of expected gestational duration, defined as the interval between LMP date and predicted due date. First, we observed that before 1990, expected gestational duration varied over the years. Women with LMP in the March–May period happened to have a longer interval to due date than women with their LMP in other months. Due date was estimated by adding the interval in month units (9 months and 7 days) to the LMP date and not days [22]. Such a method involved omitting the differences in month lengths (9 months might cover 273–276 days). This study suggests that this approach was changed around 1987, when days, instead of months, were added to the LMP to estimate due date. Second, we also observed that the expected gestational duration was longer in gestations that included the end of February during leap years. This phenomenon was due to manual estimation of gestational duration based on a widely used calendar, which does not mark an additional day in February during

leap years. Such omissions resulted in an underestimation of gestational duration, and consequently, we observed a significant increase in PTD and a decrease in post-term delivery rates during leap years.

From 1973 to 1985, there was a substantial drop in the median gestational duration (from 282 to 278 days) and post-term delivery rate (from 12% to 6%). The decreases correlated with the introduction of ultrasound-based estimation from 1982 to 1985. Studies have shown that LMP-based estimation yields a right-skewed distribution, leading to a general overestimation of gestational duration [23, 24]. Since the 1990s, when ultrasound became the most common method for estimation of gestational duration, the average rates of PTD and post-term delivery have remained stable. While ultrasound-based estimation did contribute to extra variability during the implementation period until 1992, relatively stable gestational duration has been observed since then, which proves consistency in the estimation. Until 2012, the number of post-term deliveries gradually decreased when gestational duration estimate was based on LMP date. There was also a gradual decrease in the number of post-term gestations and an increase in the number of very early (< 220 days) PTD throughout the whole study period. The findings of this study, thus, support the hypothesis that ultrasound-based estimation is more accurate than LMP-based estimation. We also show that restricting the sample to births, occurring after 1992, is desirable regarding gestational duration when using MBR data. This will avoid variability due to the simultaneous use of both estimation methods during the period when ultrasound was being introduced.

Additional variation in gestational duration might be introduced when it is stratified by mode of onset of delivery. Over the years, the accuracy of mode of onset of delivery has been improved in the MBR. In 1994, indications of whether cesarean section was performed before or after spontaneous onset of labor were introduced in the register [18]. Therefore, different definitions and coding of iatrogenic deliveries might produce different gestational duration distributions. Furthermore, changes in the definition of stillbirth also contributed to the increase in reported occurrences of PTD. In this paper, we observed an increase in the proportion of deliveries with cesarean section between 1990 and 2012 and increase in the proportion of induced onsets of delivery between 1999 and 2012.

The possibility of studying changes in pregnancy phenotype distributions and the effects of environmental factors is one of the biggest advantages of the MBR, or that of any other register-based study. Over the years, the MBR has been adjusted to improve its quality and content. In Sweden, medical diagnosis codes were switched from the ICD-8 to ICD-9 and subsequently, to ICD-10. Due to differences in the coding systems and compliance to register the codes, the quality of variables defined by ICD codes might differ. However, missing data on ICD codes, changes in reporting, and data handling from the MBR are obstacles in identifying the sources of population variability in gestational duration. This study shows that population variability of gestational duration is largely affected by changes in obstetrical practice, estimation methods, and registration methods. Better quality information will increase the accuracy of estimates obtained from both genetic and epidemiological studies with a consequent increase in statistical precision and power.

## Conclusions

The changes in data handling and obstetrical practices over the years largely contribute to the distribution of gestational duration. Digitization of medical record data entry substantially reduced the gestational duration variation due to data management. Increased variability in gestational duration might reduce precision in association studies and hamper the assessment of public health measures. We propose that future studies on gestational duration adjust the

study sample based on the findings outlined in this study; for example, accordingly to the scientific question, restrict the sample to the specific periods of time or Swedish counties.

## Supporting information

**S1 Table. Variations in the expected gestational duration among Swedish counties, Swedish Medical Birth Register, 1983–2012.** The table shows the year in which there was an observed change in the expected gestational duration. The first column shows the year in which there was a drop from the initial expected duration of 281 to 280 days. The second column shows the year in which there was a drop in the expected duration of 280 to 279 days. Sample size: n = 3 940 577. Analyses were limited to available counties. The counties where a change in the expected gestational duration was not observed during 1983–2012 (such as Uppsala and Kronoberg) are not included.
(DOCX)

## Author Contributions

**Conceptualization:** Dominika Modzelewska, Pol Sole-Navais, Anna Sandstrom, Ge Zhang, Louis J. Muglia, Christopher Flatley, Staffan Nilsson, Bo Jacobsson.

**Data curation:** Dominika Modzelewska, Bo Jacobsson.

**Formal analysis:** Dominika Modzelewska, Staffan Nilsson.

**Funding acquisition:** Bo Jacobsson.

**Investigation:** Dominika Modzelewska, Christopher Flatley, Staffan Nilsson, Bo Jacobsson.

**Methodology:** Dominika Modzelewska, Pol Sole-Navais, Anna Sandstrom, Ge Zhang, Louis J. Muglia, Christopher Flatley, Staffan Nilsson, Bo Jacobsson.

**Project administration:** Bo Jacobsson.

**Resources:** Bo Jacobsson.

**Software:** Dominika Modzelewska.

**Supervision:** Pol Sole-Navais, Anna Sandstrom, Ge Zhang, Louis J. Muglia, Christopher Flatley, Staffan Nilsson, Bo Jacobsson.

**Validation:** Pol Sole-Navais, Anna Sandstrom, Ge Zhang, Louis J. Muglia, Christopher Flatley, Staffan Nilsson, Bo Jacobsson.

**Writing – original draft:** Dominika Modzelewska.

**Writing – review & editing:** Pol Sole-Navais, Anna Sandstrom, Ge Zhang, Louis J. Muglia, Christopher Flatley, Staffan Nilsson, Bo Jacobsson.

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
