## [Decision Letter · Decision Letter 0]

29 Sep 2020

PONE-D-20-26656

Changes in data management contribute to temporal variation in gestational duration distribution in the Swedish Medical Birth Registry

PLOS ONE

Dear Dr. Modzelewska,

Thank you for submitting your manuscript to PLOS ONE. After careful consideration, we feel that it has merit but does not fully meet PLOS ONE’s publication criteria as it currently stands. Therefore, we invite you to submit a revised version of the manuscript that addresses the points raised during the review process.

There are a few minor comments from the reviewer that must be addressed.

We look forward to receiving your revised manuscript.

Kind regards,

Frank T. Spradley

Academic Editor

PLOS ONE

Reviewers' comments:

Reviewer's Responses to Questions

**Comments to the Author**

1. Is the manuscript technically sound, and do the data support the conclusions?

Reviewer #1: Yes

2. Has the statistical analysis been performed appropriately and rigorously? 

Reviewer #1: Yes

3. Have the authors made all data underlying the findings in their manuscript fully available?

Reviewer #1: Yes

4. Is the manuscript presented in an intelligible fashion and written in standard English?

Reviewer #1: Yes

5. Review Comments to the Author

Reviewer #1: Very important study showing that population variability of gestational duration is largely affected by changes in obstetrical practice, estimation methods, and registration methods. It shows eloquently the importance of increasing granularity of gestational age to days, which in Sweden is available since last century, using a combination (LMP, US) of data of the birth registry. This could be a great inspiration for other countries. Only when you overcome the binary term/preterm you can see more granular.

It also provides clever alternatives for visualizing data, the last graphic with the variation of GS in days, including leap years is one them.

The idea of having a reference for "expected gestational duration" and monitoring its trends in term period sounds very obvious but very under-explored.

Some small suggestions:

- When referring to "gestation duration mode" it would be best described by gestational duration estimates local cultures, as the author further explains?

- The effect of obstetric interventions could be a little more explored.

- A hypothesis: are ultrasound measures so uniform because there was a recalibration of growth/gestational age curves during the period?

Finally, reading it from São Paulo, Brazil, where the mean GA is 273, and for cesareans in the private sector is 268, it is great to see that in some regions in Sweden it is still 282, but at least 278. Adding granularity to GA is so feasible, so explanatory that should be widely available data. This is why I just approve the paper and I am eager to see it published.

Availability of data: under conditions.

English: very good for my non-native eyes.

6. PLOS authors have the option to publish the peer review history of their article (what does this mean?). If published, this will include your full peer review and any attached files.

Reviewer #1: **Yes: **Carmen Simone Grilo Diniz

---

## [Author Response · Author response to Decision Letter 0]

18 Oct 2020

Dear Editor Dr. Frank T. Spradley,

Thank you for considering our paper “Changes in data management contribute to temporal variation in gestational duration distribution in the Swedish Medical Birth Registry” for publication in PLOS ONE. We want to thank the reviewer for their comments on our manuscript. The detailed response to all the comments follows below. 

Yours sincerely,

on behalf of all authors,

Dominika Modzelewska

Sahlgrenska Academy, University of Gothenburg

Institute of Clinical Sciences

Dept of Obstetrics and Gynecology

SE-405 30 Gothenburg, Sweden

e-mail: dominika.modzelewska@gu.se

 

Response: We are grateful for the reviewer’s valuable comments and thank the Editor for the chance to improve our paper accordingly. The following changes were introduced to the manuscript: 1) all the headings of the major sections were adjusted, 2) all the headings of the sub-sections were adjusted accordingly, 3) whole manuscript was double-spaced, 4) references were cited before the punctuation sign, 5) short title was removed from the title page, 6) titles were removed from the author list.

Response: The following explanation was introduced to the cover letter: “The Swedish Medical Birth Registry is a national dataset; therefore, it is considered as public property. However, access to the data is given only to the researches with permission from a Swedish regional ethical review board and after approval of the research plan by the data manager. The data request may be sent to The Swedish National Board of Health and Welfare (https://www.socialstyrelsen.se/)”.

 

Reviewers' comments:

Reviewer's Responses to Questions

1. Is the manuscript technically sound, and do the data support the conclusions?

Reviewer #1: Yes

2. Has the statistical analysis been performed appropriately and rigorously? 

Reviewer #1: Yes

3. Have the authors made all data underlying the findings in their manuscript fully available?

Reviewer #1: Yes

4. Is the manuscript presented in an intelligible fashion and written in standard English?

Reviewer #1: Yes

5. Review Comments to the Author

Reviewer #1 comment #1: Very important study showing that population variability of gestational duration is largely affected by changes in obstetrical practice, estimation methods, and registration methods. It shows eloquently the importance of increasing granularity of gestational age to days, which in Sweden is available since last century, using a combination (LMP, US) of data of the birth registry. This could be a great inspiration for other countries. Only when you overcome the binary term/preterm you can see more granular.

Response: We thank the reviewer for her very supportive feedback. We also agree with and support the “granular approach” towards gestational duration. Working with binary (preterm, term) loses information and hides the stories behind the data. This is particularly the case in obstetrics where the risk curve is often parabolic in nature.

Reviewer’s comment #2: It also provides clever alternatives for visualizing data, the last graphic with the variation of GS in days, including leap years is one them.

Response: We thank the reviewer for her feedback. Communication is very important for us, and we find graphics convey a lot of information easily. We work towards better graphical data exploration and visualization. Every comment and suggestion on how to improve are very welcome.

Reviewer’s comment #3: The idea of having a reference for "expected gestational duration" and monitoring its trends in term period sounds very obvious but very under-explored.

Response: We thank the reviewer for sharing that observation. We also agree that sometimes very obvious things are left unexplored. That is why our work (research) never finishes.

Reviewer’s comment #4: Some small suggestions: When referring to "gestation duration mode" it would be best described by gestational duration estimates local cultures, as the author further explains?

Response: We understand that the reviewer is referring to Figure 3, of which, the caption includes the phrase “gestational duration mode”. The word “mode” refers to the statistical measure, the value that occurs the most frequently in the dataset. However, in gestational duration distribution we observed multiple gestational durations that occur more frequently than the neighbouring values. Such observation we called “local modes”. The phrase “gestational duration mode” refers to that distributional characteristic. 

In order to reduce the ambiguity and felicitate the understanding, we rephrased “gestational duration mode” and “local modes” to “gestational duration distribution” and “peaks”, respectively. Figure’s caption and its detailed explanation were changed: “Fig 3. Gestational duration distribution, Swedish Medical Birth Register, 1973–2012. A) Gestational duration distribution. Gestational durations that occurred at higher frequency than expected (peaks) are marked by solid black lines. Sample was reduced to pregnancies registered from 1973 to 1982, that is, to the period in which peaks were most common. Adjacent gestational durations to peaks are marked by black dashed lines. B) Average ratio of peaks’ frequency to the mean of adjacent gestational duration frequency from 1973 to 2012. Sample size: n = 3 940 577.”, lines 192-198, page 9.

Reviewer’s comment #5: The effect of obstetric interventions could be a little more explored.

Response: We thank the reviewer for inspiring further exploration of the topic. However, the aim of this paper was slightly different. In this work, we wanted to show that there are many different types of variables affecting the observations of gestational duration. The multifactorial nature of gestational duration is an obvious and often stated fact. However, we have noticed that there is a tendency in restricting the focus to biological factors. In this paper, we wanted to show that other, technical factors contribute to the observed gestational duration variability as well. Therefore, we did not intend to explore and detect all the factors contributing to the variability. We wanted to present and remind about the variability in the subgroups of the factors that affect the observations and contribute to the gestational duration variability.

Our research group explores genetic and environmental contributions to gestational duration. In our work, we see that it is very crucial to keep in mind that observed variability in gestational duration is a result of different subgroups of factors. Considering only one group of factors may lead to possible problems of interpretation. Estimates of the association studies might contain the effects of all variables. Emphasizing the complexity of the problem will inspire even more scrupulous study designs and careful interpretation of the obtained estimates.

Reviewer’s comment #6: A hypothesis: are ultrasound measures so uniform because there was a recalibration of growth/gestational age curves during the period?

Response: Over the years, gestational duration estimation approach based on the ultrasound examination has been adjusted once and that was from a report published in 2010 (1). Before 2010, the most common reference used was from Person et al 1986 (2). In 2010, Swedish Association for Obstetricians and Gynaecologists released the guidelines for pregnancy dating. In general, the adjustments relate to the timing of the ultrasound examination (first, second trimester), biometric parameters (crown–rump length, biparietal diameter), fetal sex.

Reviewer’s comment #7: Finally, reading it from São Paulo, Brazil, where the mean GA is 273, and for cesareans in the private sector is 268, it is great to see that in some regions in Sweden it is still 282, but at least 278. Adding granularity to GA is so feasible, so explanatory that should be widely available data. This is why I just approve the paper and I am eager to see it published.

Response: We thank the reviewer for your time and the approval. 

Availability of data: under conditions.

English: very good for my non-native eyes.

6. PLOS authors have the option to publish the peer review history of their article. If published, this will include your full peer review and any attached files. Do you want your identity to be public for this peer review? 

Reviewer #1: Yes: Carmen Simone Grilo Diniz

References:

 1. Kullinger M, Granfors M, Kieler H, Skalkidou A. Adherence to Swedish national pregnancy dating guidelines and management of discrepancies between pregnancy dating methods: a survey study. Reprod Health 

2. Persson PH, Weldner BM. Reliability of ultrasound fetometry in estimating gestational age in the second trimester. Acta Obstet Gynecol Scand. 1986;65(5):481–3.

---

## [Editor Report · Decision Letter 1]

23 Oct 2020

Changes in data management contribute to temporal variation in gestational duration distribution in the Swedish Medical Birth Registry

PONE-D-20-26656R1

Dear Dr. Modzelewska,

We’re pleased to inform you that your manuscript has been judged scientifically suitable for publication and will be formally accepted for publication once it meets all outstanding technical requirements.

Kind regards,

Frank T. Spradley

Academic Editor

PLOS ONE

---

## [Editor Report · Acceptance letter]

28 Oct 2020

PONE-D-20-26656R1 

Changes in data management contribute to temporal variation in gestational duration distribution in the Swedish Medical Birth Registry 

Dear Dr. Modzelewska:

I'm pleased to inform you that your manuscript has been deemed suitable for publication in PLOS ONE. Congratulations! Your manuscript is now with our production department. 

Kind regards, 

on behalf of

Dr. Frank T. Spradley 

Academic Editor

PLOS ONE